# The Influence of Artificial *Fusarium* Infection on Oat Grain Quality

**DOI:** 10.3390/microorganisms9102108

**Published:** 2021-10-07

**Authors:** Michaela Havrlentová, Svetlana Šliková, Veronika Gregusová, Bernadett Kovácsová, Andrea Lančaričová, Peter Nemeček, Jana Hendrichová, Peter Hozlár

**Affiliations:** 1Department of Biotechnologies, Faculty of Natural Sciences, University of Ss. Cyril and Methodius in Trnava, 917 01 Trnava, Slovakia; gregusova4@ucm.sk (V.G.); bernadett.kovacsova@gmail.com (B.K.); 2National Agricultural and Food Centre, Research Institute of Plant Production in Piešťany, 921 68 Piešťany, Slovakia; svetlana.slikova@nppc.sk (S.Š.); andrea.lancaricova@nppc.sk (A.L.); jana.hendrichova@nppc.sk (J.H.); 3Department of Chemistry, Faculty of Natural Sciences, University of Ss. Cyril and Methodius in Trnava, 917 01 Trnava, Slovakia; peter.nemecek@ucm.sk; 4National Agricultural and Food Centre, Research Institute of Plant Production in Piešťany, Research and Breeding Station at Vígľaš-Pstruša, 962 12 Pstruša, Slovakia; peter.hozlar@nppc.sk

**Keywords:** oat, *β*-D-glucans, fatty acids, artificial inoculation, grain quality, *Fusarium*

## Abstract

Adverse environmental conditions, such as various biotic and abiotic stresses, are the primary reason for decreased crop productivity. Oat, as one of the world’s major crops, is an important cereal in human nutrition. The aim of this work was to analyze the effect of inoculation with two species of the genus *Fusarium* on the selected qualitative parameters of oat grain intended for the food industry. Artificial inoculation caused a statistically significant decrease in the content of starch, oleic, linoleic, and α-linolenic acids in oat grains compared to the control. Moreover, artificial inoculation had no statistically significant effect on the content of *β*-D-glucans, total dietary fiber, total lipids, palmitic, stearic, and *cis*-vaccenic acids. An increase in the content of polyunsaturated fatty acids in oat grains was observed after inoculation. The most important indicator of *Fusarium* infection was the presence of the mycotoxin deoxynivalenol in the grain. The content of *β*-D-glucans, as a possible protective barrier in the cell wall, did not have a statistically significant effect on the inoculation manifestation in the grain.

## 1. Introduction

Oats (*Avena sativa* L.) belong to the main group of important cereal crops in Northern Europe and Northern America [1]. Oats are mainly used as feed for horses, but they also play an equally important role as an input material for food production and in generating pharmaceutical products and biomaterials [2]. Oats have an extensive cultivation history, playing a major role in human nutrition due to their starch and protein content, good amino acid balance [3], and their high fat content, with a relatively high ratio of unsaturated fatty acids [4,5]. Oat has a well-balanced nutritional composition. It is also a good source of total dietary fiber (TDF) and its soluble component *β*-D-glucans [3].

Oat fungal diseases are major limitations for this crop [6]. *Alternaria* and *Cladosporium* are general saprophytes or pathogens of an extensive variety of plants, and have been reported as the primary fungi in cereals worldwide [7,8]. *Puccinina coronata* f. sp. *avenae* causes crown rust disease in cultivated and wild oat (*Avena* spp.) [9]. Among the pathogenic fungi infecting oats, biotrophic pathogens, such as the powdery mildew agent *Blumeria graminis* f. sp. *avenae*, with very efficient mechanisms of spread, are the most difficult to control by means of crop management, such as rotation [6]. The main agent of root rot is *Helminthosporium sativum* [10]. *Fusarium graminearum (FG), F. culmorum (FC), F. avenaceum, F. poae*, and *Microdochium nivale* are the most common Fusarium head blight (FHB) pathogens [11]. In small grain cereals, FHB has emerged as a major problem in the Nordic European countries, and the impact of this disease in oats has been less investigated than in other cereals [12]. Taken together, these relatively few fungi can cause huge economic losses to agriculture, losses of food for consumption, and serious, often fatal diseases in humans and animals [13].

The *Fusarium* genus includes species that are widespread plant pathogens. These species cause diseases in different growing stages and parts of the plant, including the seedling blight, leaf spot, root rot, foot rot, and head blight. In cereals, FHB may be caused by at least 17 different *Fusarium* species [14]. Infection by *Fusarium* spp. is influenced by factors such as moisture and temperature, cultivar susceptibility, and cultivation practice [12]. The occurrence of the *Fusarium* species highly depends on the environment [15]. Due to changes in the climate and agricultural practices, changes in the order, as well the frequency of mycotoxins, are occurring [16]. FHB is mainly caused by *FG* and *FC*, which produce type B trichothecenes deoxynivalenol (DON) such as its acetylated derivatives, 3-acetyl-DON and 15-acetyl-DON, and nivalenol (NIV). Other species infecting cereal spikes include *F. sporotrichioides*, which produce type A trichothecenes T-2 toxin and HT-2 toxins, and *F. poae* [17]. The mycotoxin DON triggers potential cytotoxicity and genotoxicity in human peripheral blood lymphocytes via oxidative damage [18]. DON also causes abdominal distress, malaise, diarrhea, and emesis [19]. Trichothecenes are among one of the major groups of secondary metabolites produced by the species of the genus *Fusarium* and represent more than 200 compounds [20]. According to Gunnaiah and Kushalappa [21], the chemical defense against fungal pathogens, including DON produced by the *Fusarium* species, is associated with three main mechanisms of resistance: cell wall reinforcement through the deposition of lignin, the production of antimicrobial compounds, and the specific induction of defense signaling pathways. During pathogenic infection or biological elicitation, plant cells quickly respond by expressing genes encoding plant cell wall-associated non-enzymatic proteins. The cells then use the plant cell wall-associated non-enzymatic proteins to accomplish various defensive functions. These include the rapid reinforcement of cell wall strengthening against the pathogen penetration by insolubilization and the oxidative crosslinking of extensins and proline-rich proteins. In addition, the cells create a physical barrier against the pathogen invasion by clumping arabinogalactan proteins at the site of infection. The degradation of genetic materials of the pathogens by the binding of certain glycine-rich proteins with the RNA of the pathogens can be observed in the plant cell wall [22].

So far, nearly 40 identified metabolites have been reported to be associated with fatty acid metabolic pathways that may potentially affect cereal resistance against *FG*. Kachroo and Kachroo [23] summarized the importance of fatty acids and their derivatives in the defense of plants against pathogens. In addition to being essential for basal immunity and gene-mediated resistance in plants, fatty acids and their derivatives are also part of the induction of systemic acquired resistance and some of the breakdown products, such as oxylipins, and have a key role in the plant defense signaling pathway.

The oxylipin pathway is in plants a major defense signaling pathway. The oxidation of free polyunsaturated fatty acids, mainly linoleic and linolenic acid, presents the beginning of the biosynthesis of oxylipins. The action of lipoxygenases (LOX) is crucial in this step. The 9-LOXs and 13-LOXs are the main plant lipoxygenases, whereby the oxidation occurs at position 9 and 13 of the carbon chains, respectively. Two distinct biosynthetic pathways are induced. The 13-LOXs products are aimed at the formation of jasmonic acid and its derivatives. The 9-LOXs pathway leads to the production to lesser-known metabolites, which have important implications as defense factors in response to fungal attack [24]. The cultivation of oat varieties with high *β*-D-glucans content is another alternative to reduce the risk of grains contaminated by mycotoxins and thus to produce healthy and health promoting foods [25].

Oats (*Avena sativa* L.) have gained significant attention for their high content of dietary fiber, phytochemicals, and nutritional value. The consumption of oats has various health benefits, such as hypocholesterolemic and anticancerous properties [3]. The aim of the work was to analyze the influence of the *Fusarium* inoculation on the content of selected metabolites in the oat grains intended for the food industry.

## 2. Materials and Methods

### 2.1. Plant Material

Four different varieties of oats (*Avena sativa* L.) were used in the experiment. All oats were provided for research purposes by the breeder and curator of genetic resources of oats in the Slovak Republic, Peter Hozlár, Ph.D. (National Agricultural and Food Centre—Research Institute of Plant Production, Research and Breeding Station at Vígľaš-Pstruša, Detva). Ozon, Stoper, Vok, and SW Betania oats varieties were used. Ozon variety is of the hulled, yellow-grained type and originated from Germany. Ozon was registered in 2012. It is an early variety with big grains and a very good resistance to Pyrenophora leaf blotch and powdery mildew. Stoper is a hulled, yellow-grained type native to Poland, which was registered in 2003. It is a very early variety with a high-volume weight and very fine husk. Vok is a hulled variety of yellow-grained oats that originated from the Czech Republic. It was registered in 2002. Vok is a medium-late variety with fine husks and a very good resistance to crow rust, stem rust, and powdery mildew, as well as lodging. SW Betania is a spring variety of hulled, white-grained oats, and its country of origin is Sweden. The variety was registered in 2005. It is a medium-late variety with a very good resistance to crow rust, stem rust, and powdery mildew, BYDV, and lodging. Grains of oats were seeded in pots in three parallel replications. One replication was used as a control without the artificial inoculation Two replications were artificially inoculated. From each variety, 15 panicles were inoculated with *FC* and 15 inoculated with *FG*. In addition, 15 panicles of the control variant were collected. After harvest, mature grains were husked by hand and stored at −4 °C for further analysis.

### 2.2. Artificial Inoculation

Selected fungal isolates of *FC* and *FG* obtained from previous tests on cereal species were used to prepare the inoculum (data not shown). The *Fusarium* isolates were collected from commercial wheat fields in different regions of the Slovak Republic from naturally infected wheat spikes. The isolates were later preserved in the microorganism’s collection of the Research Institute of Plant Production (Piešťany, Slovak Republic). The fungal colonies of both *FC* and *FG* were cultivated on potato dextrose agar plates (PDA, Difco Lab., Detroit. MI, USA) for 21 days at 25 °C in the dark [26]. The inoculum was then prepared by covering the agar plates with sterile distilled water and soft scraping off the sporulated aerial mycelium. The hematocytometer was used to measure the concentration of inoculum which was regulated to 5 × 10^6^ conidia per ml. Approximately 1 mL of the conidial suspension was applied to each oat panicle.

For the artificial inoculation, the spraying method combined with polyethylene bag coverage was used [27,28]. The panicles were artificially inoculated at flowering using a spraying method with two separate inoculums of *FC* and *FG*. After inoculum application, the panicles were covered with polyethylene bags for 48 h.

### 2.3. Determination of Starch

The content of starch was determined using the Ewers polarimetric method (STN EN ISO 10520; 1997). The method is based on the partial hydrolysis of starch using hydrochloric acid. Then, the optical rotation of the obtained solution is measured. The optical rotation was measured using a sample cell of 200 mm optical path length at 20 °C. The final content of starch was recalculated for the dry weight of the sample.

### 2.4. Determination of TDF

The content of TDF was determined using the Total Dietary Fiber Assay procedure (Megazyme, Bray, Ireland). The method is based on the analytical methods published by Lee et al. [29] and Prosky et al. [30] (AOAC 991.43, AOAC 985.29, AACC 32-07, and AACC 32-05), and the AACC Method 32-21 and AACC method 32-06. TDF was determined in the dried samples in duplicate. The samples were incubated with heat-stable α-amylase at 100 °C to induce the hydrolysis and depolymerization of the starch. Next, protease was used at 60 °C to solubilize and depolymerize the proteins. The incubation with amyloglucosidase was used to hydrolyze the starch fragments to glucose. Samples were then treated with hot ethanol to remove the depolymerized protein and glucose from the starch and precipitate the soluble fiber. The precipitate was filtered and washed with ethanol and acetone. After drying, the precipitate was weighed. The nitrogen level was determined in one duplicate using the Dumas method (TruMac CNS Macro Analyzer, LECO Corp., St. Joseph, MI, USA). The second duplicate was used to determine the ash content. The final TDF content in the sample was then recalculated as the weight of the filtered and dried residue minus the weight of the protein and ash.

### 2.5. Determination of β-D-glucans

The total content of *β*-D-glucans in the grain was determined using the β-Glucan Assay Kit (Mixed Linkage) (Megazyme, Ireland). The streamlined method has been successfully evaluated by the AOAC (Method 995.16) and the AACC (Method 32-23), and published by McCleary and Codd [31]. Milled mature oat grains were suspended and hydrated in a sodium phosphate buffer (pH 6.5), incubated with purified lichenase, and filtered. Then, an aliquot of the filtrate was hydrolyzed with *β*-glucosidase. Oxidase/peroxidase reagent was used to examine the produced glucose. The content of the *β*-D-glucans was recalculated on the dry matter and determined as the mean of three replications as a percentage.

### 2.6. Determination of Lipids and Fatty Acids

The method by Soxhlet according to the standard (STN 461011-28) was used in duplicate to determine the total lipids content. To evaluate the fatty acids composition, methylesters were prepared according to Christoperson and Glass [32]. Next, they were analyzed by GC-MS (7890B GC system, 5977A MSD, Agilent Technologies, Santa Clara, CA, USA) using the capillary column HP-5MS ultra inert (30 m × 250 µm × 0.25 µm) and MS detector. The temperature gradient was used according to the following conditions: Column temperature program: initial temperature 150 °C (held for 4 min.), then increased to 230 °C (held for 5 min.) and finally increased to 280 °C (maintained 19 min.). The parameters of the capillary column inlet were as follows: temperature: 250 °C, pressure: 10.8 psi, total helium pressure: 169.32 mL/min., split ratio: 200:1. HP-5 ms ultra-inert column: dimensions 30 m × 250 μm × 0.25 μm, initial temperature: 150 °C, flow rate: 0.82746 mL/min., pressure: 10.8 psi, average value: 34 613 cm/s. The sample was injected at the volume 1 µL.

The MS parameters were as follows: MSD Transfer-line temperature: 280 °C, quadrupole temperature: 150 °C, ion source temperature: 230 °C, electron energy: 70 eV, record full mass spectra (SCAN type), scanning range: 50–550 *m*/*z*, scan speed: 1.562 (N = 2), gain factor: 1. Fatty acids were identified using the authentic standards of the C4–C24 fatty acids methyl esters mixture (Supelco, Bellefonte, PA, USA), and the mass spectrum of samples was compared with the spectrum in Library NIST2007 in ChemStation 10.1 (Agilent Technologies). All fatty acids were quantified as the area percentages of the fatty acids in the chromatogram.

### 2.7. Quantification of DON

#### 2.7.1. Preparation of Standard Solution of DON

The standard DON solution in acetonitrile was prepared in a concentration of 1 mg·mL^−1^ and stored at −18 °C. The calibration standards of DON were prepared by diluting the standard solution in the mobile phase prepared from acetonitrile and deionized water (10:90, *v*/*v*).

#### 2.7.2. Preparation of the Sample

A volume of 20 mL of the mobile phase was added to 5 g of the ground grain sample. Then, the solution was stirred in an orbital shaker at 220 rpm for 1 h. The mixture was centrifuged at 2500 rpm for 30 min. Then, 4 mL of filtrate was applied to the immunoaffinity column.

#### 2.7.3. Clean-Up on Immunoaffinity Columns (IAC)

The IAC Donprep column (R-Biopharm Rhône, Glasgow, UK) was used for DON isolation. At first, the columns were conditioned, with the filling solution being present at each IAC. A volume of 4 mL of extract of the sample, as described by the pre-treatment procedures, was passed through the column by gravity or slight vacuum, if appropriate. The column was washed with 10 mL of deionized water, and the washing eluates were discarded. After drying the column under vacuum for 10 min, DON was eluted using 2 mL of methanol. The elution solvent was subsequently dispersed on a rotary evaporator at a water bath at 50 °C. The obtained residue was dissolved in the mobile phase and analyzed by HPLC.

#### 2.7.4. HPLC Procedures

Analyses were performed on an HPLC system of Agilent Technologies 1200 Series, equipped with an auto-sampler and the analytical column Zorbax SB-C18, 4.6 × 250 mm i.d. with the sorbent particle size of 5 μm, connected to a guard column Zorbax SB-C18, 12.5 × 4.6 mm i.d., with the sorbent particle size of 5 μm (both from Agilent Technologies). A diode-array detector (DAD) was set at following wavelengths: sample/bandwidth: 220/16 nm, reference/bandwidth: 360/80 nm. As a mobile phase, the mixture of acetonitrile: deionized water (10:90, *v*/*v*) was used at a flow rate of 1 mL·min^−1^. The injected volume of the sample was 50 μL and was performed by the auto-sampler. The column temperature was adjusted to 25 °C and was automatically controlled by the Agilent ChemStation (Agilent Technologies). The final concentration of DON in the oat samples was calculated using the equation C [μg·kg^−1^] = C*·F (1000/m), where C* is the concentration of the mycotoxin estimated from the calibration curve in μg·ml^−1^ (injection concentration); F is the conversion factor, which includes the initial and final volumes of the sample taken into analysis, as well as the factor 1000, which presents a conversion from μg·g^−1^ to μg·kg^−1^; and m is the sample weight in grams.

### 2.8. Statistical Evaluation

The results obtained from the chemical analyses were statistically analyzed using the JMP 11.0 software. The following statistical methods were used: correlation analysis (CA), analysis of variance (ANOVA) with post-hoc test (Duncan), and principal component analysis (PCA).

## 3. Results

In our experiment, the content of primary metabolites (starch, TDF, *β*-D-glucans, total lipids with the profile of fatty acids), as well as the concentration of DON in the oat grain were analyzed. With the analysis, we aimed to observe the changes in the oat grain composition after the inoculation with the *Fusarium* species. The average content of starch in the analyzed oat grains in the control variant was 41.50%, with a range from 39.51% (SW Betania) to 43.65% (Stoper). After the artificial inoculation, a decrease was detected in the average content of starch. The average content of starch after the *FC* inoculation and *FG* inoculation was 32.76% and 37.37%, respectively, showing a reduction of 21% and 10%. All analyzed oat varieties showed a uniform statistically significant decreasing trend after the inoculation. Significant differences in the starch content were also detected between the *FC* and *FG* artificial inoculation (Table 1).

In the content of TDF, an increase was observed after the artificial inoculation compared to the control, although it was not statistically significant. In the control, the average content of TDF was 25.48%, ranging from 23.47% (Stoper) to 27.02% (Ozon). After the inoculation with *FC* and *FG*, the average content of TDF was 29.30% and 27.52%, respectively, showing an increase of 4% and 8% (Table 1).

The artificial *Fusarium* inoculation influenced the content of *β*-D-glucans, but this was not statistically significant. In the control variant, the average content of this cell wall polysaccharide was 3.49%, where 3.13% (Ozon) and 4.17% (Stoper) were the marginal amounts. After the inoculation with *FC*, a decrease was detected, with an average content of 2.91% and a reduction of 17%. In all analyzed varieties except SW Betania, a reduction was observed. After the inoculation with *FG*, an increase in the content of *β*-D-glucans was detected (3.53%), although in Stoper and Vok, a reduction in the content of this polysaccharide was observed (Table 1).

The average content of total lipids was 4.06%, with a range of 3.66% (Ozon) and 4.50% (SW Betania). After the artificial inoculation, a statistically not significant decrease was observed. After the *FC* inoculation, to the average content of total lipids decreased to 3.60%, and after the *FG* inoculation, it decreased to 3.72%, showing a reduction of 11% and 8%, respectively. All analyzed oat samples except Stoper showed the trend of a reduction in the total lipids content compared to the control (Table 1).

The presence of DON in the oat grain is an indicator of *FC* and *FG* infection. In our experiment, the average concentration of DON in the control oat grains was 157.03 µg·kg^−1^, ranging from 0.00 µg·kg^−1^ (Ozon) to 378.76 µg·kg^−1^ (SW Betania). After the inoculation, a large statistically significant increase was observed in the mycotoxin’s concentration. Statistically significant differences were also observed between *FC* and *FG* artificial inoculation. After the *FC* inoculation and *FG* inoculation, the average concentration showed an increase of 2096.01 µg·kg^−1^ and 1317.18 µg·kg^−1^, respectively (Table 1).

The profile of fatty acids was analyzed in our experiment in the oat samples of the control variant and after the artificial inoculation with the *Fusarium* species. Linoleic, oleic, and palmitic fatty acids were detected in proportions exceeding 15% of the total lipids. Minor fatty acids in the oat samples were observed, including *α*-linolenic, stearic, and *cis*-vaccenic (Table 2).

The levels of linoleic, palmitic, *α*-linolenic, stearic, and *cis*-vaccenic acids increased after the *Fusarium* artificial inoculation. However, the effect of the inoculation was only statistically significant in the case of linoleic and *α*-linolenic acids. The level of oleic acid, on the other hand, decreased statistically significant after the inoculation compared to the control (Table 2).

The average level of linoleic acid in the control samples was 39.50% of the total lipids. The average level of linoleic acid increased to 41.58% of the total lipids after the *FC* inoculation and to 41.84% after the *FG* inoculation, showing an increase of 6% and 6.5%, respectively. The increase was observed in all analyzed oat samples. In the control oat samples, the average level of palmitic acid was 16.65% of the total lipids. The average level of palmitic acid increased to 17.10% and 16.85% after the artificial inoculation with *FC* and *FG*, respectively. The level of α-linolenic acid increased from 1.91% in the control samples to 2.72% and 2.18% after the artificial inoculation with *FC* and *FG*, respectively, showing an increase of 42% and 14%, respectively. In the control sample, the average level of stearic acid was 1.85%, and it also increased after the *Fusarium* inoculation. After the *FC* inoculation, the average levels of stearic acid experienced a mild increase to 1.86%. After the *FG* inoculation, the increase was stronger, with average levels increasing to 2.01%. The levels increased by 0.5% and 9%, respectively. In addition, the level of *cis*-vaccenic acid increased after the artificial inoculation with the *Fusarium* species. In the control samples, it was 1.09%. After the *FC* inoculation, it raised mildly to 1.10%, and after the *FG* inoculation, it increased markedly to 1.33%. In contrast, the level of oleic acid decreased after the artificial *Fusarium* inoculation. In the control samples, the level of oleic acid was, on average, 39.19%. After the *FC* inoculation and *FG* inoculation, it decreased to 35.80% and 35.81%, respectively, showing an identically reduction of 8% (Table 2).

Figure 1 shows the statistical analyses of the results. Figure 1A represents the grouping of the analyzed samples in terms of the differences or similarity based on the grain quality parameters. The group of control samples (marked with an empty circle), together with the artificially inoculated variety Stoper, formed one common cluster, which indicates the lowest rate of DON accumulation in Stoper after the *FC* and *FG* artificial inoculation. The second cluster consists of varieties that responded more strongly to the *FC* and *FG* inoculation by accumulating higher concentrations of DON compared to the first cluster. It is visible that the inoculated varieties Ozon and Vok showed the lowest degree of tolerance to the *FC* inoculation. In comparison, the Vok (inoculated with *FG*) and SW Betania showed a lower DON accumulation in grain (Figure 1A).

In Figure 1B, the PCA shows clusters of samples based on their similarity, as well as the relationships between parameters and their effect on the principal components. The control variants showed a higher starch content compared to the inoculated variants. Higher contents of starch, total lipids, and oleic acid were observed in the varieties Stoper and SW Betania. A negative correlation coefficient (r = −0.901; *p* < 0.001) between the starch content of the control samples and the DON concentration was observed. A positive correlation coefficient (r = 0.798; *p* < 0.001) between the total lipid content and the oleic acid portion indicates that the higher the lipid content in the grain, the higher the content of the given fatty acid after the artificial inoculation with the pathogenic microorganisms. Another positive correlation coefficient (r = 0.516; *p* < 0.010) was detected between the DON concentration and the linoleic acid portion, and between the DON concentration and the *α*-linolenic acid portion (r = 0.468; *p* < 0.050), which indicates that the higher the concentration of DON in the grain, the higher the portion of individual fatty acid in oat lipids. Generally, an increase in the portion of polyunsaturated fatty acids was observed after the artificial inoculation compared to the control (Figure 1B).

The ANOVA test, which was used to statistically evaluate the measured data in a more detailed way, showed the most statistically significant differences in the concentration of DON after the artificial inoculation (as factor) with both *Fusarium* species compared to the control variants. It is clear from the results that there is a statistically significant difference in the concentration of DON between the control variants and the individual inoculated samples. Statistically significant differences can also be observed in the starch content between the control and inoculated variants. In addition, statistically significant differences were detected in the levels of linoleic acid, oleic acid, α-linolenic acid, total lipids, and DON between the control and inoculated variants. According to these results, it can be established that the concentration of the mycotoxin DON is the most statistically important indicator of *Fusarium* inoculation. From the results, it can also be concluded that, in comparison with other studied oat varieties, the oat variety Stoper showed the most statistically significant differences in the measured quality parameters between the control and inoculated variants. The obtained results also indicate that, in the case of *β*-D-glucans content, there were no statistically significant differences between the control and inoculated variants.

## 4. Discussion

In our experiment, the content of starch in the analyzed oat grains of the control variant ranged from 39.51% to 43.65%, with an average content of 41.50%. Our results are in accordance with the literature. For example, Sterna et al. [33] defined an average content of 48.08% in the grains of hulled oat. After the artificial inoculation with the *FC* and *FG*, a decrease in the content of starch was observed. It is likely that the pathogenic microorganism used this storage polysaccharide as a source of energy by penetrating to the plant. In the work of Wang et al. [34], the *FC* infection had no analytically detectable influence on the starch and the total insoluble dietary fiber content of the wheat grain, although microscopy revealed obvious damage to the starch granules in the infected samples. In addition, tested thermostable characteristics suggested that the *α*-amylose of *FC* may damage starch granules [34].

Rakić et al. [35] reported that, in the oat grain, amylose is integrated into the fractions of lipids and proteins. In addition, between the lipid and amylose content in oat starch, a positive correlation has been detected [36,37]. In oat starch, higher lipid content (>1.36%) can increase the hydrophobic properties and thus increase the stability of the starch film [38].

Our control oat samples contained, on average, 25.48% of TDF. In the literature, similar content was reported [33]. No uniform trend was observed in our experiment with the artificial *Fusarium* inoculation, although an increase was observed in Ozon after both the *FC* and *FG* inoculations, and in all other varieties after the *FC* inoculation. This increase might have been caused by the decrease of starch weight, which increased the proportion of TDF in the oat grain. Moreover, it could be due to the attack of the pathogen, as the grain did not sufficiently strengthen its cell wall. As reported by Rashid et al. [22], when plant cells are attacked by a pathogen, the cell wall proteins are used by the cells to perform several defense functions against the pathogen penetration. The mechanism of action involves binding glycine-rich proteins to the RNA of the pathogens at the site of infection to degrade the genetic materials of the pathogenic microorganisms. High-protein oat varieties tend to attach less often by fusariosis, accumulate lower concentrations of toxins, and better adapt to biotic stress [39].

In our experiment, the content of *β*-D-glucans in the control samples was, on average,3.49%, with contents ranging from 3.13% to 4.17%. In the literature [25,33,40], similar amounts have been published. However, differences between naked and hulled oat grains were detected, with statistically significant higher contents in naked grains [25]. From our results, it is not possible to observe a uniform trend in the content of this cell wall polysaccharide after the artificial inoculation. The plant metabolism could have either a positive or a negative effect on the expression of pathogens in the plant [39], which has been confirmed by analyzing oat varieties with different degrees of resistance to FHB [6,12]. After the attack of the pathogen, the levels of individual metabolites vary in infected plants according to the plant resistance, the presence of pathogen-produced mycotoxins, as well as the level of pathogenic DNA in grains [25,41]. At the microbial attack, callose—(1 → 3)-*β*-D-glucan—is synthesized and stored in the papillae [42]. This polysaccharide contributes to the formation of a barrier, which slows down the penetrating pathogen. Following this, it is assumed that (1 → 3)(1 → 4)-*β*-D-glucans in the cell wall of the selected cereals will also form a physical barrier to defend the inner cell from the pathogenic microorganism [25,43]. Therefore, based on our experiment, we suppose that oat varieties with high content of *β*-D-glucans (Stoper) could accumulate less DON in the grains. In our experiment, Stoper contained 4.17% of *β*-D-glucans in the control variant. Compared to the other varieties (3.13–3.50%), the content of *β*-D-glucans was higher in the Stoper variety. Moreover, the variety was more susceptible to the *FC* and *FG* inoculation, as it accumulated less DON in the grain after the *Fusarium* inoculation. Such putative protective role of *β*-D-glucans was described also by Havrlentová et al. [25] and Martin et al. [43]. Higher *β*-D-glucans content might enhance the natural resistance of the plant against the fungal pathogen by contributing to the type V resistance (resistance against toxins accumulation) caused by the antioxidant activity of *β*-D-glucans [43,44]. A possible explanation for the reduction of *β*-D-glucans in inoculated plants is the fact that *β*-glucosidase has a role in the degradation of *β*-D-glucans in plant cells. This enzyme is activated during several physiological processes in the plant cell, as well as during the protection of plants against pathogens [45].

In our experiment, the average content of total lipids in the oat grains of the control variant was 4.06%, with the range of 3.66–4.50%. The literature describes the content ranging from 5% to 6% [33] or from 4.8% to 6.0% in dehulled oat grains [46]. The infection of plants by pathogenic microorganism is connected with the lipid metabolism [47]. The reason for the decrease in the lipid content in inoculated oat samples may be attributed to the increased activity of oxylipins [48,49] as a tool suppressing the pathogenic expression of the *Fusarium* species [24].

The presence of the mycotoxin DON in oat grain is a symptom of the infection. In the control samples, the average concentration of DON was 157.03 µg·kg^−1^, and after the inoculation, a large increase was observed. According to the literature, the DON disrupts the normal function of plant cells at the molecular level by inhibiting proteosynthesis, which affects cell signaling, differentiation, and proliferation [50]. The DON production is associated with mechanisms of resistance, such as cell wall reinforcement realized through the lignin deposition, antimicrobial compounds production, and the induction of defense signaling pathways [21]. DON is required for spread of the pathogen because it allows the invading fungus to expand through the rachis from the infected to the adjacent spikelet [47]. The mycotoxin has the function of a translation inhibitor, thereby suppressing the formation of cell wall thickening at the rachis node and inhibiting an important response of the host [51,52,53]. The production of DON by a pathogen triggers several signals in the plant, such as specific metabolites that already exist in a healthy plant or are induced in response to a pathogenic attack. It has been confirmed that the plant responds actively to the attack. Boedi et al. [47] reported some functional categories in the plant that were significantly overrepresented in the upregulated active-plant gene set associated with cellular defense and detoxification, as well as the lipid and secondary metabolites metabolism.

In oat oil, oleic (C18: 1), linoleic (C18: 2), and palmitic acid (C16: 0) are the major fatty acids [33,46], accounting for approximately 90–95% of all fatty acids [54]. The minor fatty acids observed in our study, as well as in the literature [55,56], were *α*-linolenic, stearic, and *cis*-vaccenic. The composition of fatty acids in oat oil is important from a technological [56] as well as a nutritional point of view [46,57].

In all fatty acids except the oleic acid, an increase was observed after the artificial inoculation. The portion of the oleic acid, on the other hand, decreased after the inoculation compared to the control, in accordance with the findings of Wang et al. [58] in wheat (*Triticum aestivum* L.) after the *Fusarium* inoculation. A possible explanation for the reduction in oleic acid content is the fact that unsaturated fatty acids (oleic, linoleic, and *α*-linolenic acids) have antimicrobial effects capable of limiting the growth of pathogenic microorganisms, including *FG* [59]. Another explanation is that phenylpropanoids and phenols in cuticular wax or cutin could provide some protection against fungi attacks, as *FC* and *FG* and oat cuticles could be enhanced by the incorporation of fatty acids [60].

Palmitic acid, the main saturated fatty acid in oat oil, increases the stability of oat oil and protects fatty acids against peroxidation [57]. The increase of palmitic acid was observed not only in our study, but also in the study of Wang et al. [58], where the concentration increased by 9% in infected wheat varieties compared to the controls. The increase is probably an indicator of fungal lipase activity, although this is not a very early phenomenon in the infection process [61].

In general, an increase in the content of polyunsaturated fatty acids was observed after the inoculation, in agreement with the literature [23,58,62]. The results obtained by studies performed on inoculated wheat plants showed that the profile of fatty acids changed due to the growth of microscopic fungi [23,58]. Fatty acids and their derivatives play an important role in the plant’s defense against biotic stress. They are essential for basal immunity and plant resistance genes [23]. The modification in the synthesis of fatty acids is a result of the presence of filamentous fungi and their metabolism [58], and the increase in selected fatty acids concentration could be due to the activity of fungal lipases [61].

## 5. Conclusions

Oat is a cereal of great importance for human nutrition. Therefore, the grain must be healthy and of high nutritional value with high portions of TDF, *β*-D-glucans, and unsaturated fatty acids. The inoculation with *FC* and *FG* caused a statistically significant decrease in the content of starch and oleic, linoleic, and α-linolenic fatty acids in oat grains compared to the control. The artificial inoculation had no significant effect on the content of *β*-D-glucans, TDF, total lipids, palmitic, stearic, and *cis*-vaccenic fatty acids. Generally, the content of polyunsaturated fatty acids increased after the artificial inoculation with the *Fusarium* species. In our experiment, oat varieties better tolerated the *FG* inoculation. On the contrary, the *FC* inoculation appeared to be stronger, producing a higher concentration of DON in the grain compared to *FG*, whereby the presence of the mycotoxin DON is the most important indicator of the inoculation. A negative correlation between the starch content and the DON concentration (r = −0.901; *p* < 0.001) was observed. In addition to the content of starch, the portion of linoleic acid in the oat oil was the quality parameter most influenced by the fungal inoculation. The content of *β*-D-glucans, as a putative protective barrier in the cell wall of the oat grain, did not have a statistically significant effect on the manifestation of the *Fusarium* infection in the oat grain. On the other hand, Stoper, with the highest content of this polysaccharide, accumulated the lowest concentrations of DON in the grain after the artificial *Fusarium* inoculation.

## Figures and Tables

**Figure 1 microorganisms-09-02108-f001:**
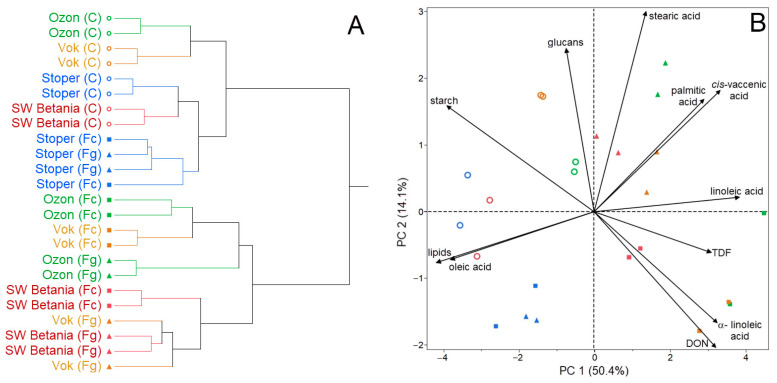
Cluster analysis (**A**) and principal component analysis (PCA) (**B**) of the content of starch, *β*-D-glucans, total lipids, TDF, stearic acid, palmitic acid, *cis*-vaccenic acid, linoleic acid, oleic acid, *α*-linolenic acid, and DON concentration in the oat grains as a manifestation of the artificial inoculation with *FC* and *FG***.** Empty ring (C) means control. The *FC* and *FG* inoculations are marked by the square and triangle, respectively.

**Table 1 microorganisms-09-02108-t001:** The content of the quality parameters in the oat grains in the control conditions and after the artificial inoculation expressed as the mean ± standard deviation. Among the columns, the different letters indicate statistically significant differences according to the Duncan test (*p* ≤ 0.05).

	Genotype	Starch (%)	TDF (%)	*β*-D-Glucans (%)	Total Lipids (%)	DON (µg·kg^−1^)
Control	Ozon	40.59 ± 0.08 ^a^	27.02 ± 0.66 ^a^	3.13 ± 0.05 ^a^	3.66 ± 0.17 ^a^	0.00 ± 0.00 ^a^
	Stoper	43.65 ± 0.62 ^a^	23.47 ± 0.11 ^a^	4.17 ± 0.04 ^a^	4.02 ± 0.02 ^a^	187.50 ± 15.91 ^a^
	Vok	42.25 ± 0.25 ^a^	25.13 ± 0.66 ^a^	3.50 ± 0.00 ^a^	4.08 ± 0.20 ^a^	61.88 ± 2.65 ^a^
	SW Betania	39.51 ± 0.26 ^a^	26.31 ± 0.93 ^a^	3.16 ± 0.03 ^a^	4.50 ± 0.05 ^a^	378.76 ± 29.17 ^a^
Mean		41.50	25.53	3.49	4.29	157.03
*FC* inoculation	Ozon	30.82 ± 0.34 ^b^	27.87 ± 4.63 ^a^	2.66 ± 0.06 ^a^	2.93 ± 0.17 ^a^	1949.41 ± 189.40 ^b^
	Stoper	37.76 ± 0.95 ^b^	27.06 ± 0.24 ^a^	3.89 ± 0.39 ^a^	4.35 ± 0.05 ^a^	1469.91 ± 9.82 ^b^
	Vok	30.99 ± 0.00 ^b^	32.73 ± 0.14 ^a^	1.67 ± 0.04 ^a^	3.39 ± 0.00 ^a^	2339.72 ± 3.80 ^b^
	SW Betania	31.48 ± 0.17 ^b^	29.54 ± 0.62 ^a^	3.42 ± 0.12 ^a^	3.72 ± 0.35 ^a^	2625.00 ± 7.58 ^b^
Mean		32.76	29.30	2.91	3.60	2096.01
*FG* inoculation	Ozon	34.90 ± 0.26 ^c^	29.88 ± 0.54 ^a^	6.87 ± 0.25 ^a^	3.11 ± 0.06 ^a^	1207.47 ± 91.53 ^c^
	Stoper	39.43 ± 0.00 ^c^	26.31 ± 0.26 ^a^	2.16 ± 0.01 ^a^	4.30 ± 0.01 ^a^	990.19 ± 3.11 ^c^
	Vok	36.81 ± 0.35 ^c^	28.11 ± 4.23 ^a^	1.51 ± 0.25 ^a^	3.51 ± 0.39 ^a^	1609.28 ± 0.00 ^c^
	SW Betania	38.34 ± 0.17 ^c^	25.77 ± 0.07 ^a^	3.60 ± 0.21 ^a^	3.95 ± 0.45 ^a^	1461.79 ± 6.28 ^c^
Mean		37.37	27.52	3.54	3.72	1317.18

TDF: total dietary fiber; DON: deoxynivalenol.

**Table 2 microorganisms-09-02108-t002:** The level of monitored fatty acids in the oat grains in the control conditions and after the artificial inoculation expressed as the mean ± standard deviation. Among the columns, the different letters indicate statistically significant differences according to the Duncan test (*p* ≤ 0.05).

	Genotype	Linoleic Acid (%)	Oleic Acid (%)	Palmitic Acid (%)	*A*-linolenic Acid (%)	Stearic Acid (%)	*C**is*-vaccenic Acid (%)
Control	Ozon	40.97 ± 0.08 ^a^	36.72 ± 0.23 ^a^	16.95 ± 0.01 ^a^	2.22 ± 0.06 ^a^	1.76 ± 0.11 ^a^	1.40 ± 0.30 ^a^
	Stoper	37.69 ± 0.20 ^a^	41.77 ± 0.21 ^a^	15.86 ± 0.16 ^a^	1.98 ± 0.01 ^a^	1.70 ± 0.18 ^a^	1.01 ± 0.06 ^a^
	Vok	40.41 ± 0.49 ^a^	36.94 ± 0.62 ^a^	17.72 ± 0.28 ^a^	1.84 ± 0.13 ^a^	2.18 ± 0.15 ^a^	0.93 ± 0.13 ^a^
	SW Betania	38.14 ± 0.10 ^a^	41.33 ± 0.21 ^a^	16.10 ± 0.32 ^a^	1.63 ± 0.02 ^a^	1.77 ± 0.26 ^a^	1.05 ± 0.19 ^a^
Mean		39.30	39.19	16.66	1.92	1.85	1.09
*FC* inoculation	Ozon	44.25 ± 0.17 ^b^	31.14 ± 0.64 ^b^	17.80 ± 0.23 ^a^	3.55 ± 0.22 ^b^	1.83 ± 0.42 ^a^	1.46 ± 0.04 ^a^
	Stoper	39.58 ± 0.09 ^b^	39.85 ± 0.44 ^b^	15.57 ± 0.77 ^a^	2.24 ± 0.07 ^b^	1.82 ± 0.12 ^a^	0.51 ± 0.58 ^a^
	Vok	42.65 ± 1.34 ^b^	33.89 ± 1.61 ^b^	17.46 ± 0.24 ^a^	2.88 ± 0.08 ^b^	1.78 ± 0.01 ^a^	1.35 ± 0.11 ^a^
	SW Betania	39.86 ± 1.70 ^b^	38.33 ± 0.28 ^b^	17.57 ± 0.05 ^a^	2.22 ± 0.02 ^b^	2.04 ± 0.06 ^a^	1.08 ± 0.02 ^a^
Mean		41.59	35.80	17.10	2.72	1.86	1.10
*FG* inoculation	Ozon	41.27 ± 0.29 ^b^	35.25 ± 0.05 ^b^	17.12 ± 0.07 ^a^	2.60 ± 0.12 ^a^	2.21 ± 0.21 ^a^	1.58 ± 0.08 ^a^
	Stoper	40.41 ± 0.35 ^b^	39.00 ± 0.15 ^b^	15.58 ± 0.65 ^a^	2.39 ± 0.13 ^a^	1.58 ± 0.28 ^a^	1.07 ± 0.25 ^a^
	Vok	42.93 ± 0.53 ^b^	34.19 ± 0.09 ^b^	17.35 ± 0.23 ^a^	1.99 ± 0.01 ^a^	2.25 ± 0.28 ^a^	1.32 ± 0.06 ^a^
	SW Betania	42.77 ± 1.69 ^b^	34.80 ± 0.84 ^b^	17.33 ± 0.68 ^a^	1.77 ± 0.09 ^a^	2.01 ± 0.12 ^a^	1.35 ± 0.04 ^a^
Mean		41.85	35.81	16.85	2.18	2.01	1.33

## Data Availability

Not applicable.

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
