# Peer review of "The Influence of Artificial Fusarium Infection on Oat Grain Quality"

_microorganisms, 2021, doi:10.3390/microorganisms9102108_

Round 1

Reviewer 1 Report

All comments, questions and corrections are marked in PDF file.

Author Response

I would like to thank the reviewer very much for the energy, time and comments he took to improve the quality of the manuscript. In the text in pdf format all the comments and remarks from the reviewer were observed and I made changed in the text by accepting all of them. I deleted the required words or parts of sentences, I corrected tables 1 and 2, I corrected the cited ideas, I added new publication and cited it in Disccussion... All the changes are seen in the atteched pdf document. Thank you once again. 

Reviewer 2 Report

The manuscript entitled “The influence of artificial Fusarium infection on oat grain quality” reports a study on four selected varieties of oat grains, showing the effect of artificial inoculation with Fusarium graminearum and F. culmorum species on some important qualitative parameters and on deoxynivalenol (DON) content in oat grains. The fungal inoculation with both the Fusarium species led to a heavy increase of DON, more evident with F. culmorum inoculation. At the same time, general increase of polyunsaturated fatty acids, except oleic acid, was observed after fungal inoculation, suggesting a role in plant defence against fungi.
This study provides useful information on the effects of two important mycotoxigenic fungi and DON on nutritional component of oats, a widely cultivated cereal crop and consumed in human diet.
The manuscript is quite well written and exposed. The background is well organized, the methods are appropriate and results are well reported and discussed.

Below are reported in detail some suggestions to be revised:

While I think it is right to use the acronym FG and FC for Fusarium graminearum and F. culmorum, respectively, but I think that the name of genus Fusarium should be written in full when it is alone, in lines: 46, 50, 64, 66, 99, 130, 239, 251, 275, 333, 340, 398, 411, 439, 469, 477, 479.

Provide a picture with higher resolution for Figure 1a, since it is no possible to read the names.

The results of ANOVA analysis reported in lines 331-345 should be shown also in a Table.

Line 45. Delete “F.” 
Lines 50-51. Replace the part of the sentence with: “Fusarium genus includes species which are common plant pathogens and cause diseases in different growing stages and parts of the plant: seedling……”
Line 57. Invert the position of the words “mainly” and “caused”
Line 58. Replace “-“ with “, such as”
Lines 59-60. Replace the sentence with: “Other species infecting cereal spikes include F. sporotrichioides, which produce type-A trichothecenes T-2 and HT-2 toxins, and F. poae [17].”
Lines 71-77. The sentence is too much long. Rearrange it.
Line 86. Replace “chiefly” with “mainly”.
Line 107. Add “oats varieties” after “Betania”. Delete “variety of”
Line 109. Delete “the year”
Lines 110-111. Replace the sentence with “Stoper is a hulled, yellow-grained oats native to Poland and registered in 2003.”
Line 121. Delete “ones”
Line 126-127. Replace “in the flowering phase” with “at flowering”
Lines 125-128. Move the entire part after “5x106 propagules per ml.” (line 138)
Line 129. Replace “tested” with “previous tests on”
Line 130. Add “(data not shown)” after “inoculum”
Line 200. It is not recommended to begin a sentence with a number. I suggest to add “A volume of” before “20 ml”.
Line 201. Delete the point (.) after “h”
Line 242. I suggest to write that in brackets was shown the percentage of reduction. For example, you could replace the sentence with “The average content of starch was 32.76% after FC inoculation and 37.37% after FG, showing reduction of 21 and 10%, respectively.”. The same is in other parts of the Results section.
Line 251. Move “in our experiment” after “inoculation”.
Lines 264-265. Replace “F. infection” with “FG and FC infections” and replace “In our experiment with the artificial inoculation, in oat grains of the control the average concentration of DON...” with “In our experiment the average concentration of DON in control oat grains…”
Line 270. Add “of” before values.
Line 275-276. Replace the sentence “The main fatty acids with the proportion more than 15% of total lipids were detected the fatty acids linoleic, oleic, and palmitic.” With “Linoleic, oleic, and palmitic fatty acids were detected in proportion more than 15% of total lipids.”
Lines 304-307. The sentence is not clear. Rewrite.
Line 310. Replace immunity with tolerance
Line 311. Delete “the”
Line 314. Delete “It is seen from the results, that”
Line 324. Replace “what” with “which”
Line 356. Replace “F.” with “FG and FC”. Do the same in lines 412 and 464.
Line 364. Delete “the fact,” and add commas after “that” and after “amylose”
Lines 370-371. Delete comma after “both” and delete “the” before “FC”
Line 391. Replace “According the fact, there is an assumption that” with “According this, it is assumed that”
Line 392. Delete “will”
Lines 393-395. Replace the entire sentence with “Therefore, based on our experiment, we suppose that oat varieties with high content of β-D-glucans (Stoper) could accumulate less DON in the grains.
Line 417. Delete “by F. species”
Line 430-431. Delete the entire sentence from “The main” to “detected”
Line 436. Replace “excepting” with “except”
Line 437. Delete “statistically significant”
Line 438. Replace “compared to the control what was described also by Wang” with “compared to the control, in accordance with Wang”
Line 442. Replace “A possible” with “Another”; delete “the fact”
Line 443. Delete “presented”
Line 444. Replace “of” with “as”
Line 453. Replace “what is in acceptance” with “agreeing” and replace “of model” with “obtained by”
Line 455. Move “the profile of fatty acids changed” after “showed that in line 454
Lines 462-464. The sentence is not clear. Rewrite.
Lines 470-471. Replace “inoculation and the FC…” with “inoculation; on the contrary, FC…”. Delete brackets.
Line 475. Replace “the inoculation” with “fungal inoculation”

Author Response

I would like to thank the reviewer for very usefull comments and remarks. All of them I supplemented to the text. Only one remark regarding the ANOVA results I could not, what I explained in the attached document. 

Once again, thank you very much for the time, energy and usefull comments. 

Round 2

Reviewer 1 Report

The manuscript has been thoroughly improved in accordance with my comments and can be published in the present form.